

# Baseflow in karst regions is significantly higher than the global average and exhibits spatial variability

Author :Ze Yuan [a,b], Qiuwen Zhou [a,b]*, Yuan Li [a,b]*, Yuluan Zhao [a,b], Shengtian Yang [c,d]

Affiliation:

a. School of Geography and Environmental Science, Guizhou Normal University, 550001 Guiyang, China

b. Karst Ecosystem Field Scientific Observation and Research Station of Guizhou Normal University & Guanling Autonomous County, 561300 Guanling, China

c. Institute of Ecological Civilization, Guizhou Normal University, 550001 Guiyang, China

d. College of Water Sciences, Beijing Normal University, 100875 Beijing, China

Corresponding author: Qiuwen Zhou *, Yuan Li*

Email: zqw@gznu.edu.cn, Yuan Li: liyuan7pro@163.com



Abstract: The distinct hydrogeological configurations of karst terrains engender
fundamentally divergent baseflow regimes compared with non-karst systems. However,
there is still some uncertainty in the understanding of baseflow in global karst regions
due to the variability of methods and differences in natural conditions in different
regions. In this study, runoff data from 1375 karst basins around the world were
summarized, and graphical and digital filtering methods were used to estimate baseflow
in global karst regions and to analyze their spatial differences and trends. The results
show that the baseflow index of global karst areas is about $78 \pm 6.9\%$, which is
significantly higher than the global average baseflow index (60%). The baseflow index
of karst regions in different climatic zones also differed significantly, in which the
average baseflow index of arid karst regions (82%) was significantly higher than the
average baseflow index of subtropical karst regions (77%). Even within the same
climate zone, the base flow index of different regions may also have significant
differences, and the difference of some regions is even >10%. Vegetation factors
reflected in primary productivity have the highest influence on baseflow in karst regions
(15%), while climatic factors (relative humidity, air temperature, etc.) have a lower
influence on BFIs in karst regions (less than 5%). From the time series trend, the global
karst baseflow index shows an increasing trend, about 1.5% from 1960 to 2015. These
results help us to further understand karst hydrological processes and the response
mechanism of karst hydrology under climate change.

Key words: Baseflow; Karst; Hydrographic graphical method; climate zone; global
runoff data; hydrogeology



## 1. Introduction

Baseflow plays a central role as a slow recharge component of groundwater to runoff as a hydrological stabiliser (Mukherjee et al., 2018; Chen et al.,2019). The proportion and dynamic characteristics of baseflow in runoff not only regulate the ecological balance threshold of rivers, but also profoundly affect the resilience of watersheds in response to climate fluctuations (Saedi et al ., 2022; Hare et al., 2021; Yang et al., 2023). Therefore, accurate quantification of the characteristics of baseflow can help to understand the runoff evolution pattern and its response mechanism to regional environmental changes (Mei et al., 2024; Kuehne et al., 2023).

Recent studies on baseflow estimation have revealed its spatial variability characteristics. Among them, Xie et al (2023), based on a coupled analysis of baseflow separation and climate models for 15,000 catchments worldwide, pointed out that the average contribution of baseflow to river runoff was about 60%. However, there are significant regional differences under this macroscopic pattern, e.g., baseflow index (BFI) calculations by Beck et al (2013) for 3,394 watersheds globally show that BFI is generally higher in tropical and temperate-cold regions than in arid and semi-arid zones (e.g., North and South Africa, Central Asia, and Australia). Regional scale studies further refine the spatial differentiation pattern, such as the United States, where the BFI is higher in the east than in the west, India, where the BFI is higher in the east than in the west, and the Yellow River basin, where it is higher in the upstream and downstream and lower in the middle reaches, whereas the BFI of the Wei River basin in the Loess Plateau shows a gradual decrease from the upstream to the downstream (Mei et al ., 2024 ; Sharma and Mujumdar ., 2024 ; Lyu et al ., 2023 ; Zhang et al ., 2019).

The current study characterises global baseflow features, but the unique hydrological structure of karst landscapes (e.g., pipes and fissures) makes the baseflow features obtained from the above study significantly less applicable in karst regions (Jiang et al., 2024 ; Ford & Williams, 2007). The current study found significant regional differences in BFI characteristics in karst regions around the world. In particular, the high permeability of karst media in tropical karst regions (e.g., Sumatra, Java) contributes to the rapid conversion of precipitation to groundwater, as analysed in three sub-basins of the Brantas Hulu watershed, where the BFI exceeds 80% (Pratama and Adji., 2020), and the study of three basins in Jonggrangan area also showed BFI of more than 87 per cent (Khomsiati et al., 2021). Seasonal differences in



BFI are highlighted in subtropical karst regions (Mediterranean Sea, Southern China),
such as central Italian basins with baseflow contributions spanning 30-76%, rising to
88-90% in dry months (Longobardi and Loon., 2017), and southeastern France has
significant differences in baseflow contributions (27%-61%) in years of abundant and
dry water (Guisiano et al., 2024). Temperate karst regions such as the Sierra Nevada
karst region in North America generally have BFI higher than 65% (Tobin and
Schwartz., 2019). The BFI in the karst region of southwest China is 57% (Mo et al.,
2025), a stable BFI of ≥55% in temperate karst in central Ireland (Foran et al., 2021),
and a BFI of $36\pm10\%$ in the karst mountains of eastern China (Lyu et al., 2022).
In summary, studies of baseflow in karst regions have revealed their obvious
spatial heterogeneity. A large number of studies have characterised the baseflow
characteristics of karst under different climatic zones, and also outlined the regional
baseflow characteristics of karst under different climatic zones (Tagne and Dowling.,
2018). However, existing studies still have obvious limitations, starting with an over-
focus on localised features in small regions, such as watershed studies in southern China
and the Mediterranean (Guisiano et al., 2024; Mo and Ruan., 2021), which makes the
results of the study not necessarily representative of the global karst region. The second
is the variability of research methods, such as hydrographic methods (graphical
methods, digital filtering methods), isotope tracer methods, etc. (He et al., 2019; Yang
et al ., 2021 ; Arnold et al.,2013). The difference in focus of the different methods also
reduces the commonality of the findings. These two reasons have led to a lack of
characterisation of overall features and reasonable quantification of regional differences,
despite the exploration of baseflow characteristics of karst basins in different regions
of the world (Wu et al., 2017; Mei et al., 2024). Therefore, the complete characterisation
of baseflow in the global karst region using reasonable methods and the accurate
quantification of the overall characteristics and regional differences of baseflow in the
global karst region are still urgently needed.
The aim of this study is to explore the baseflow characteristics and their internal
differences in the global karst region and to discuss the influence of different factors on
baseflow in karst regions. Global public runoff data were selected for the study, and
daily-scale runoff data from 1375 watersheds within the karst region were selected.
Twelve baseflow separation methods, including four graphical methods and eight
digital filtering methods, were used to separate the baseflow from the runoff data and
calculate BFIs. The reliability of the results was assessed using the Kling-Gupta





Efficiency (KGE) (Gupta et al., 2009) and Nash-Sutcliffe Efficiency (NSE) (Nash and
Sutcliffe., 1970) coefficients, and finally, the XGBoost model was used to analyse the
influencing factors of the 12 indices on baseflow.
## 2.Materials and methods
## 2.1 Data sources
### 2.1.1 Runoff data
We have selected regions with a concentrated distribution of karst landscapes
worldwide. And combined with global watershed data (Lehner and Grill, 2013),
Köppen climate zoning, and urban distribution, select runoff observation stations with
less human activity and watershed areas less than 2500 km$^2$. Thus daily runoff data for
1412 watersheds with different time spans have been selected. The runoff data mainly
comes from the Global Runoff Data Center (https://www.bafg.de/GRDC), The
European Water Archive (https://ne-friend.bafg.de/servlet/), National River Flow
Archive, UK (https://nrfaapps.ceh.ac.uk/nrfa/nrfa-api.html), Brazilian National Water
Authority (https://zenodo.org ), The National Hydrological Data Archive of Canada
(https://wateroffice.ec.gc.ca/), The Chinese Ministry of Water Resources
(http://www.cjh.com.cn/), The National Hydrological Information System of the United
States (https://waterdata.usgs.gov/nwis).
Due to quality differences in data from different hydrological observation stations,
it is necessary to clean the data from these 1412 stations. Exclude sites with severe data
gaps and supplement data from sites with a small amount of missing data. We use cubic
spline interpolation and linear interpolation to supplement data with missing amounts
less than 30 days. Finally, daily runoff data of 1375 watersheds in different time ranges
of karst regions worldwide were obtained. This includes 221 watersheds in tropical
karst zones, 91 watersheds in arid karst zones, 490 watersheds in subtropical karst zones,
and 568 watersheds in temperate karst zones (Figure 1).



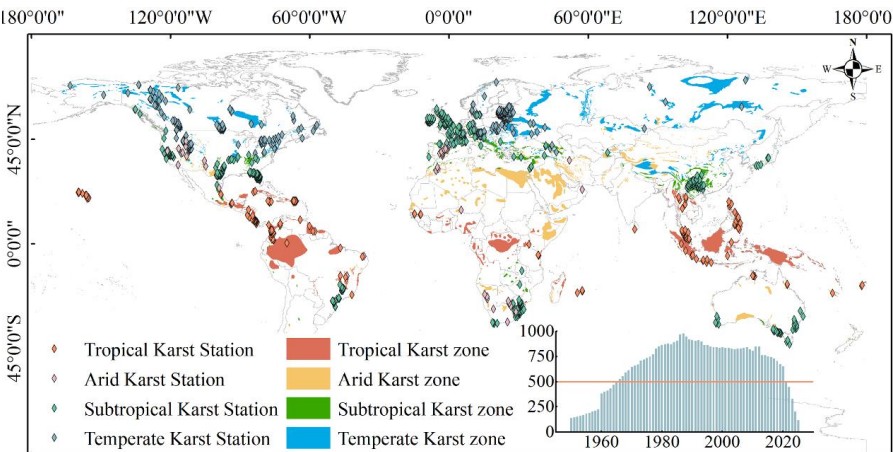

Figure 1. The distribution of karst landscapes and hydrological stations in various climate zones around the world. The bar chart represents the number of hydrological stations selected in each year, with the vertical axis indicating the number of selected hydrological stations and the horizontal axis indicating the year. We selected years with over 500 hydrological stations that meet the requirements within the same year for subsequent analysis.

## 2.1.2 Selection of potential influencing factors of base flow

In order to analyse the influencing factors of baseflow, we further selected daily-scale runoff data from 744 hydrological stations during 2011-2012 out of the 1375 hydrological stations mentioned earlier to calculate baseflow. The purpose of further selecting the hydrological stations is to ensure the continuity of the data while at the same time ensuring that the stations can cover the major karst regions of the world. We selected a total of 12 potential influences. Climatic factors included temperature and rainfall, and geological factors included depth to bedrock, water storage in epikarst, slope, elevation, and soil evaporation. Other factors included runoff, population density, gross primary productivity (GPP), relative humidity, and surface radiation, for a total of 12 factors (Table 1).

Table 1. Detailed information on the 12 influencing factors

| Name | Temporal scale | Spatial resolution | Data sources |
|---|---|---|---|
| Runoff volume | Monthly everage | - | The same as the runoff data in Section 2.1.1 |
| Epikarst water storage volume | Monthly everage | 30 arc-second | GES(Goddard Earth Sciences)DISC(Li et al .,2019) |
| Bedrock depth | - | 0.25km×0.25 km | ISRIC — World Soil Information (Hengl et al .,2017) |



| Air temperature | Monthly everage | 30 arc-second | |
|---|---|---|---|
| Precipitation | Monthly everage | 30 arc-second | Climatic Research Unit gridded Time Series(arris et al .,2020) |
| Relative humidity | Monthly everage | 0.1°×0.1° | |
| Elevation | - | 30 arc-second | Worldclim(Fick and Hijmans .,2017) |
| Slope stepness | - | 30 arc-second | |
| Available soil moisture | multi-year average | 1km×1km | HWSD(Harmonized World Soil Database)(Wieder et al .,2014) |
| Population density | multi-year average | 30 arc-second | LandScan Global 30 Arcsecond Annual Global Gridded Population Datasets (Bright et al., 2013) |
| Gross primary production | multi-year average | 0.25°×0.25° | TU Data Repository(Wild et al.,2022) |
| Land-surface radiation | Monthly everage | 10km | Data Center of the Qinghai-Tibet Plateau(Tang .,2019) |


## 2.2 Methods

### 2.2.1 Baseflow separation methods

Commonly used methods for baseflow separation include isotope tracer methods,
hydrological modelling methods and hydrographic methods (including graphical
methods and digital filtering methods). However, the isotope tracer method relies on
high-precision isotope data and is difficult to be extended in data-poor areas, while the
hydrological modelling method is limited by the empirical nature of the parameters as
well as the regional nature. Therefore, considering the characteristics of the study area
(wide range and insufficient observational data), we chose the hydrographic method,
which requires less data and is relatively simple.
The computational tool used for baseflow separation in this study is from the
Python library baseflow (https://pypi.org/project/baseflow) developed by the team of
Xiaomang Liu at the Chinese Academy of Sciences, which contains four graphical
methods and eight digital filtering methods that allow simultaneous implementation of
multiple methods for baseflow separation (Xie et al. 2024). In addition to this the
baseflow library evaluates each method when separating the baseflow and obtains an
optimal method. In this study, the baseflow library was used to separate baseflow from
global runoff data and calculate its multi-year average BFI (Figure 6).
Graphical methods are techniques for isolating baseflow by analysing runoff
hydro-graph. The four graphical methods used in this study are Fixed Interval Method



(FIM), Local Minimum Method (LMM), Sliding Window Method (SW) and UK
Institute of Hydrology (UKIH).

180        Digital filtering is a baseflow segmentation method that uses digital signal

processing techniques to separate baseflow from runoff by designing specific filters.
These methods usually involve one or two parameters, such as the recession coefficient.
The recession constant is automatically estimated in the baseflow library using the
Brutsaert method, and the second parameter is calibrated using the multi-objective
optimisation method proposed by Arnold (Brutsaert., 2008; Rammal et al., 2018). The
methods used in this work include the Boughton Method (Boughton), Chapman-
Maxwell Filter Method (CM), Chapman Filter Method (Chapman), Exponential
Weighted Moving Average (EWMA), Eckhardt Filter Method (Eckhardt), Furey Digital
Filter Method (Furey), Lyne-Hollick Digital Filter Method (LH), and Willems Digital
Filter Method (Willems).
2.2.2 Evaluation metrics for baseflow separation methods

192        In order to validate the accuracy of different baseflow separation methods in

karstic regions, we chose two metrics, KGE and NSE coefficients, to measure the
effectiveness of different methods in separating baseflows. The methodology used by
Xie et al (2020) for measuring and assessing the effectiveness of baseflow separation
methods in the US region was used, which centred on screening for strict baseflow
points.
2.2.3 Attributional analysis methods

199        Due to the significant differences in magnitude of the potential influences selected

at the global scale (a few hydrological stations are at extremely high elevations, whose
actual differences are compressed after normalisation, making it difficult to adequately
characterise the effect of elevation on baseflow), traditional linear models or distance
metric-based algorithms are susceptible to magnitude interference. Therefore, we chose
the magnitude-insensitive XGBoost model, which naturally circumvents the feature
scale difference problem through the splitting rule of the tree structure (Niazkar et
al.,2024; Zhang et al., 2022). In addition, the model's built-in regularisation mechanism
and subsampling strategy can effectively suppress overfitting and guarantee the model's
generalisation ability in complex geographic data. The model also supports parallel
computing with automatic processing of missing values, which significantly improves
the computational efficiency of large-scale spatial datasets (Chen and Guestrin.,2016).
3.Results

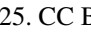


## 3.1 Validation of the applicability of baseflow separation methods

We counted the best separation methods filtered in the Baseflow library for baseflow separation for each hydrological station data. From the results in Fig. 2, 28% of the hydrological stations are suitable for baseflow separation using the graphical method, 71% of the stations are suitable for baseflow separation using the digital filter method, and 1% of the stations have no obvious suitable separation method. Among them, the EWMA method is the most effective for baseflow separation in karst area, with 24% of hydrological stations suitable for baseflow separation, followed by the Eckhardt method, with 21% of hydrological stations suitable for baseflow separation.

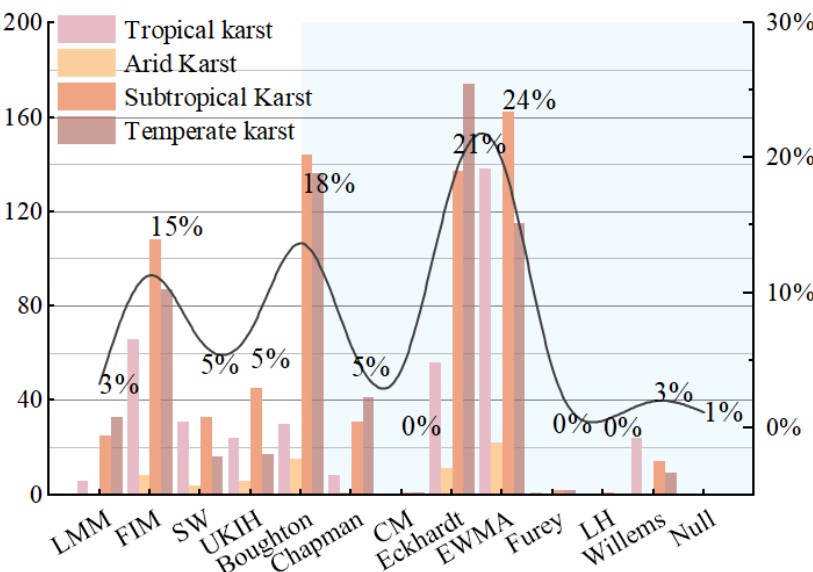

Figure 2. Percentage of best separation methods in the karst region and number of best separation methods in each climatic zone. Graphical methods are shown within the white background, digital filtering methods are shown within the light blue background, and different coloured bars correspond to different climatic zones.The X-axis shows the 12 baseflow separation methods, the Y-axis (left) shows the number of hydrological stations, and the Y-axis (right) shows the number of hydrological stations covered by each of the optimal baseflow separation methods as a proportion of the number of all hydrological stations, which corresponds to the black curve.

Figure 3(a) shows the KGE coefficient distributions of different methods, from the results, some digital filtering methods (orange) have concentrated KGE coefficient distributions and the values are close to 1. For example, the five methods, Boughton, Eckhardt, EWMA, Furey, and Willems, which indicate that the applicability of these



five base-flow separation methods is high and effective in the karst region. The KGE
coefficients of the graphical method (green) are also well distributed, with most of the
KGE coefficient distribution ranges greater than 0.5 and the average KGE coefficients
of each method are greater than 0.75. It indicates that the graphical method also has
high applicability in the karst region. On the other hand, the three digital filtering
methods of Chapman, CM and LH have discrete distributions from the distribution of
KGE coefficients, although their average values are all greater than 0.5. It indicates that
the results obtained by these three methods are more fluctuating when dealing with data
from different hydrological stations, and it also shows that these three methods are less
stable when performing baseflow separation.

244       The distribution pattern of the NSE coefficients of the different methods in Fig.

3(b) is similar to that of Fig. 3(a). The NSE coefficients of the five methods, Boughton,
Eckhardt, EWMA, Furey, and Willems, have a concentrated distribution and high mean
values, which further suggests that these five methods are effective in separating the
baseflow in karst regions. The distribution of NSE coefficients of the four graphical
methods (in green) is generally stable although the range of NSE coefficients increases
compared to the KGE coefficients, and their mean values are all greater than 0.5. The
distribution of NSE coefficients of the three digital filtering methods of Chapman, CM,
and LH is still more discrete (-0.5 to 1), which further indicates that the applicability of
these methods in karstic regions is low.

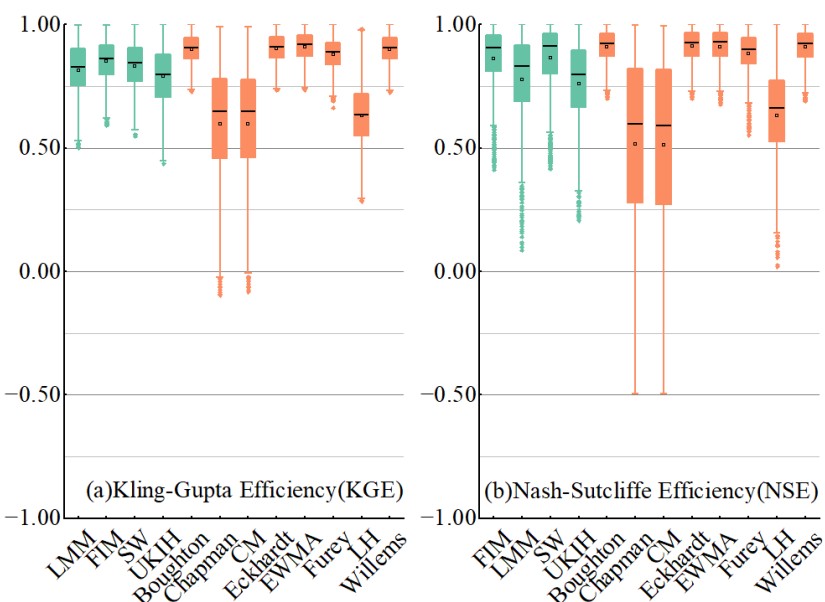






Figure 3. KGE coefficients (a) versus NSE coefficients (b) for 12 baseflow separation
methods. The X-axis indicates each separation method and the Y-axis indicates the
value of the coefficients. The green color in the graph indicates the graphical method
and the orange color indicates the digital filtering method. The black lines within the
boxplot indicate the mean values, with upper and lower limits of 1.5 times Interquartile
Range(IQR), and exceeding the range is considered as an outlier, which is labeled in
the form of dots at the top and bottom of the boxplot.
From the distribution characteristics of KGE and NSE coefficients in different
climatic zones (Figure 4), the KGE coefficients of multiple separation methods in
tropical karst have discrete distributions, with CM and Chapman ranging from -1.5 to
1. The NSE coefficients are similar to those of the KGE, but with a relatively centralised
distribution. The distribution of coefficients of graphical methods in the arid karst
region are all discrete, and the digital filtering method is still the CM and Chapman
methods presenting a low concentration. The distribution of KGE coefficients in
subtropical and temperate karst is relatively stable and concentrated, and the overall
distribution of KGE coefficients of Chapman and CM are also discrete, while the KGE
coefficients of FIM and SW are close to 1, which indicates that these methods are more
effective in separating the baseflow in subtropical and temperate karst regions.
According to Figures 2 and 3, considering the high KGE and NSE coefficients and
the number of most suitable hydrological stations, we selected four more suitable
methods for baseflow separation in karst regions, which are one graphical method (FIM)
and three digital filtering methods (Boughton, Eckhardt, EWMA).

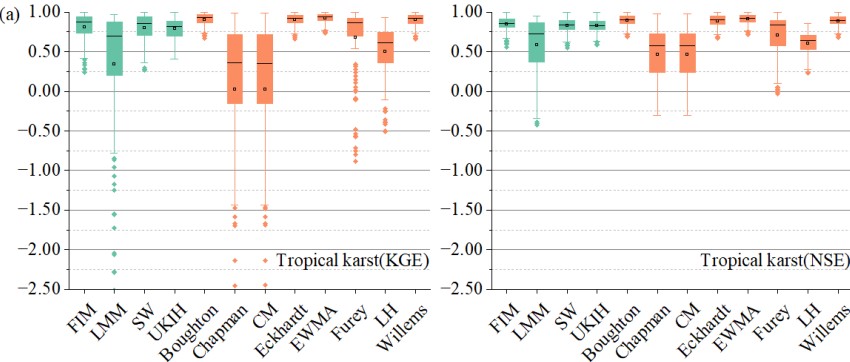




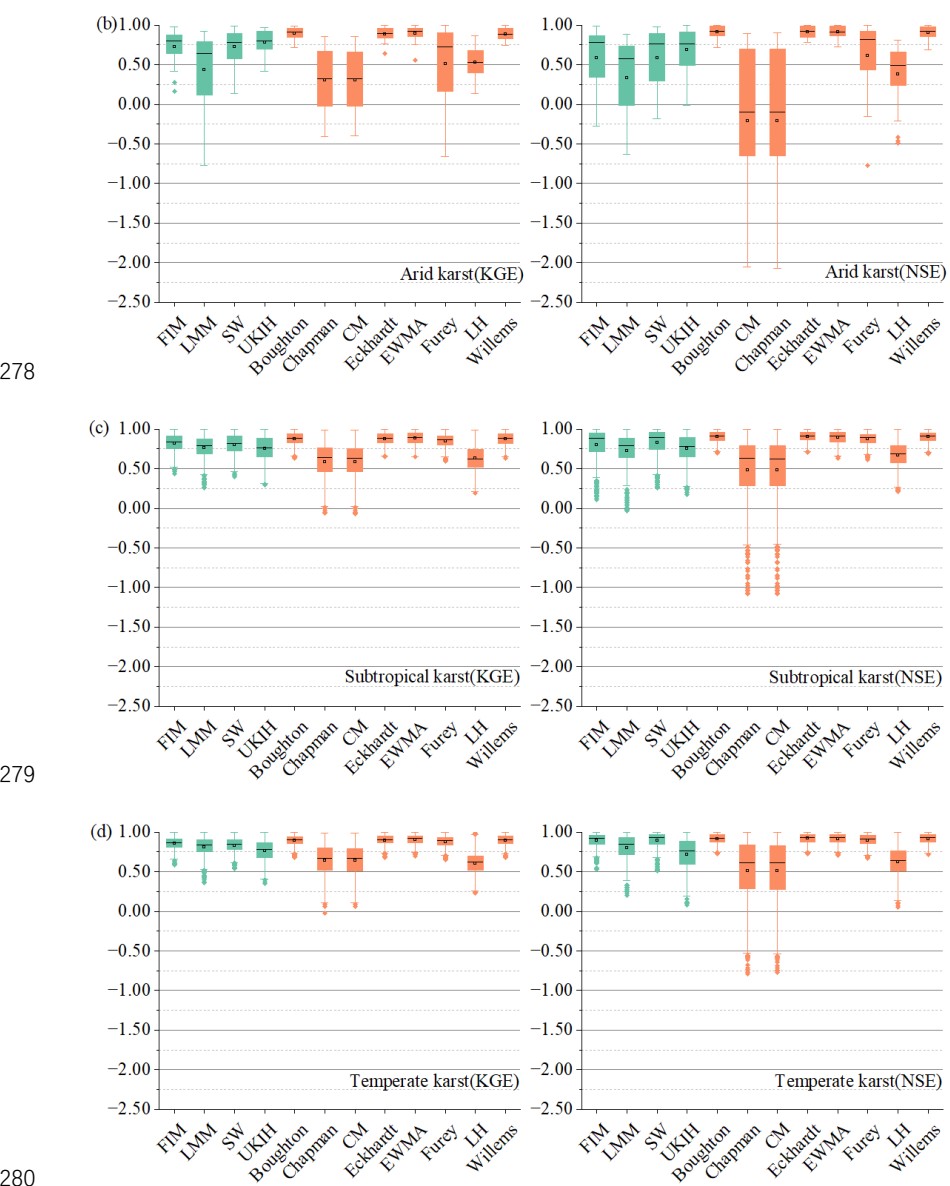




Figure 4. KGE coefficients (left column) versus NSE coefficients (right column) for karst regions in different climatic zones (labeled at the bottom right of each component Figures). The X-axis indicates each separation method and the Y-axis indicates the value of the coefficients. The green color in the graph indicates the graphical method and the orange color indicates the digital filtering method. The black lines within the boxplot indicate the mean values, with upper and lower limits of 1.5 times IQR, and



exceeding the range is considered as an outlier, which is labeled in the form of dots at
the top and bottom of the boxplot.
## 3.2 Differences in baseflow indices obtained by different methods
over time
From Figure 5a, it can be found that the four graphical methods have different
effects on baseflow separation in karst regions. Among them, the BFIs derived by FIM
and SW are similar, with an average value of about 86%. Moreover, the BFI shows an
increasing trend of low amplitude with the year, with low fluctuation degree and high
stability. The mean value of BFI derived from LMM is about 83%, and the trend of
change with years shows a decreasing and then increasing trend, while the result of
UKIH method is low, with a mean value of about 77%, and its BFI also shows a slow
increasing trend with years.
The results in Figure 5b can be found that although there are differences in the base
flow indices obtained by different digital filtering methods, most of the methods obtain
similar base flow indices and have similar trends with respect to year. In contrast, the
results of the two methods Chapman and CM differ significantly from those of the other
six methods. The mean value of the BFI obtained by the two methods is about 58%,
and there is a small decrease followed by a slow increase in the trend.

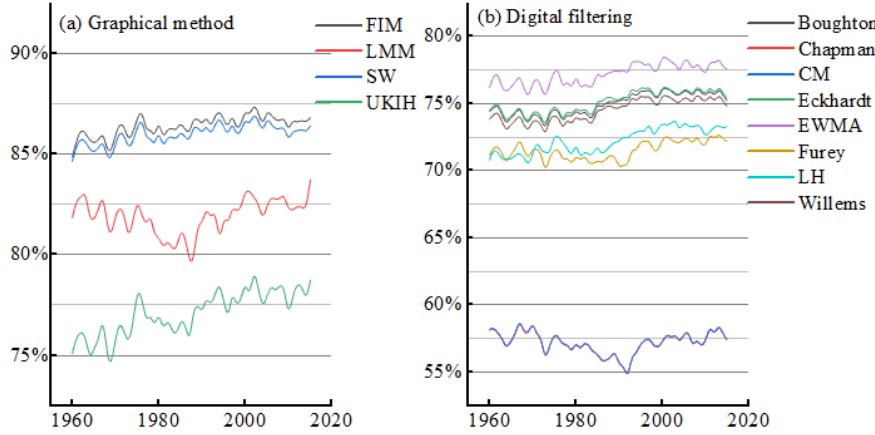


Figure 5. Global BFIs for karst regions calculated by the 12 baseflow separation
methods, with the x-axis indicating the year and the y-axis the BFI.
In order to analyze the reasons for the differences between these two methods (CM
and Chapman) and other methods in separating baseflows, we selected one hydrological
station in each climatic zone and generated baseflow curves obtained by the different
methods in different climatic zones (Figure 6). Since the CM method is an improvement





of Chapman by adding a maximum baseflow limit to the Chapman method, and its
internal mechanism is consistent, Chapman was used as a proxy. In addition, the
Eckhardt method with high KGE and NSE coefficients is chosen as a comparison. From
Figure 6, we find that when runoff increases, the Eckhardt method can respond quickly
and baseflow increases rapidly, while the Chapman method responds to the increase in
runoff to a lesser extent and by a lower amount than Eckhardt. Overall, Chapman
responds more slowly to the recharge of precipitation than the other methods, and this
feature also makes the Chapman method less discriminating for baseflow compared to
the other methods.

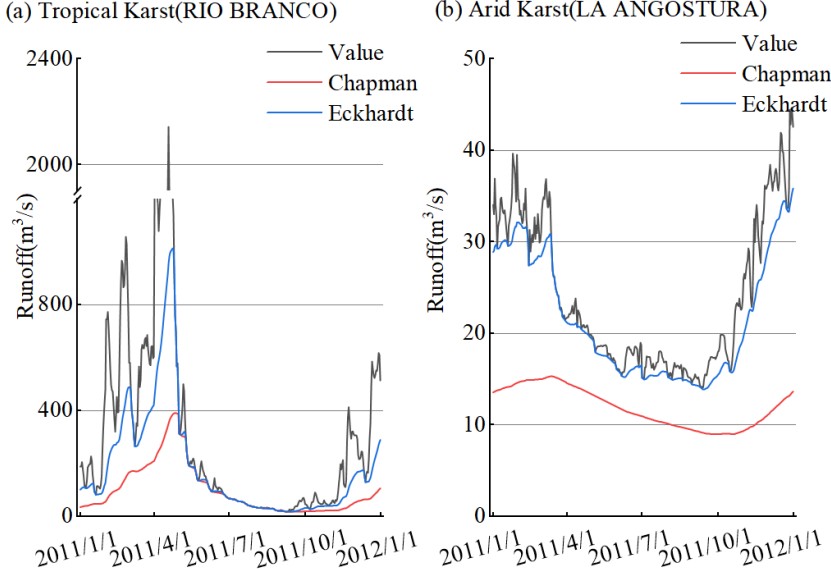


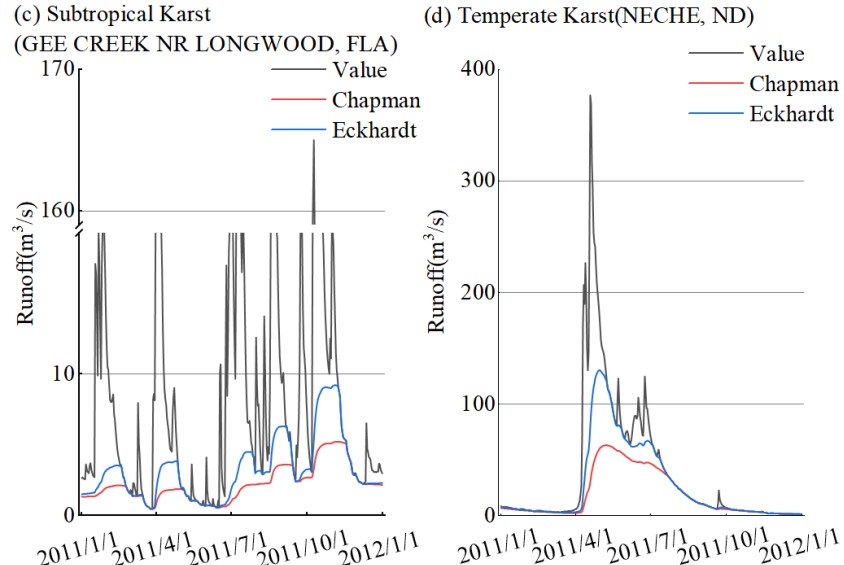

Figure 6. baseflow curves for different climatic zones (Eckhardt and Chapman methods were chosen as representatives), where the X-axis represents time and the Y-axis represents runoff. The black curve (Value) represents the runoff volume. Names of hydrological stations are in parentheses.

## 3.3 Global base flow characteristics

In order to more clearly characterize the BFI in karst basins, we calculated the BFI in non-karst basins globally using the same method. Figure 7 shows that BFIs in karst basins are significantly higher than in non-karst basins. The BFI of karst basins is $78\pm$ 6.9%, while the BFI of non-karst basins is about 60%. This indicates that baseflow in karst basins is significantly underestimated if only global average conditions are considered and baseflow in karst basins is not calculated separately.

As can be seen from Figure 7, there are differences in the characteristics of BFIs over time in different climatic zones. The BFI in the tropical karst region generally shows an increasing trend. From 1960 to about 1990, the base flow index in tropical karst showed an increasing trend, and since 1990 the base flow index remained at about 80% and then stabilized. The BFI in arid karst region is the highest, with a mean value of about 85%. In general, the BFI in arid karst region shows a decreasing trend, and the annual mean BFI fluctuates greatly, with poor stability. The BFI of subtropical karst region is more stable, always maintained at about 78%. The characteristics of BFI in temperate karst regions are similar to those of tropical karst, showing a slow increase and remaining stable at around 80%.





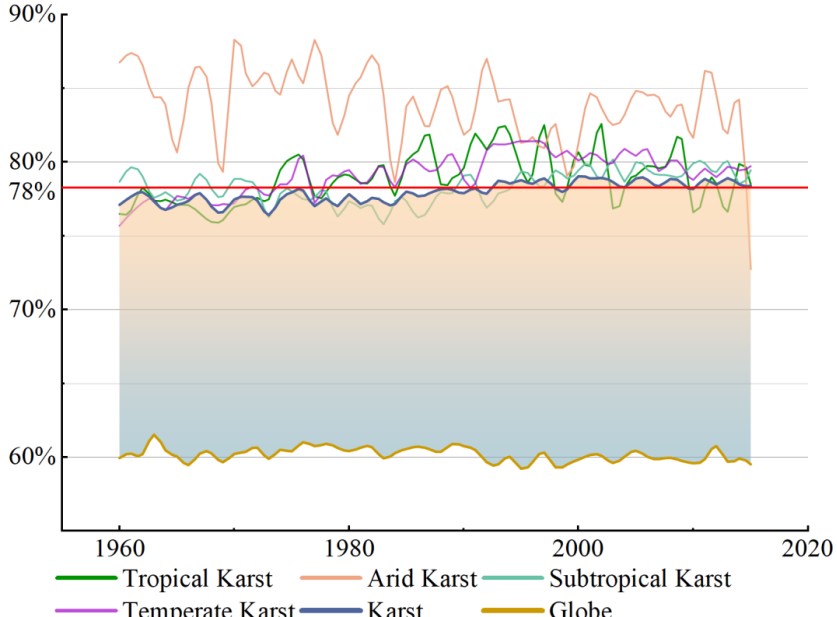

344

Figure 7. Annual BFIs for each region of global karst versus global annual BFIs. x-axis
indicates the year, y-axis indicates the BFI, and the red straight line is the overall mean
of the BFI. The orange curve at the bottom indicates the global BFI, and the dark blue
line indicates the BFI for global karst regions.

The average of BFIs obtained by the four methods (FIM, Boughton, Eckhardt,
EWMA) was used as the BFI for the global karst region and linearly regressed against
the year (Figure 8). The results show an increasing trend in the BFI in the global karst
region, with an increase of about 1.5% from 1960 to 2015. One of the obvious increase
periods is from 1980 to 2000. Since 2000, the BFI in the global karst region has
stabilised, fluctuating in the range of 78.5% $\pm$ 0.5%.



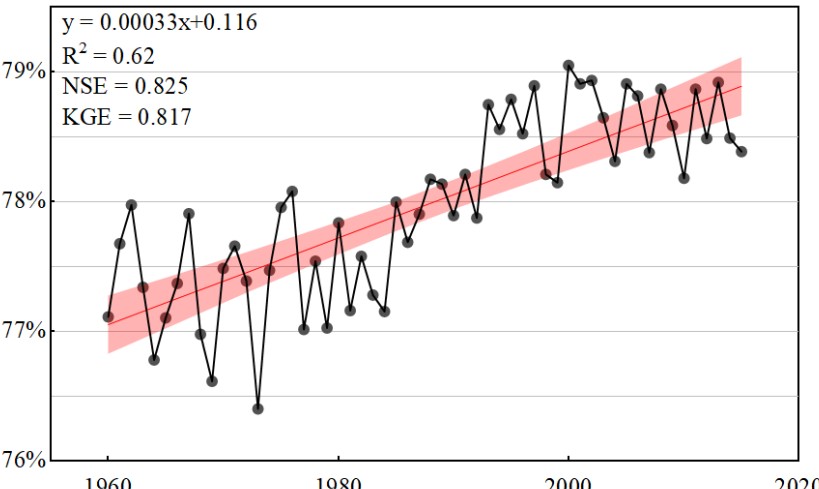

Figure 8. Annual mean BFIs over time for global karst regions. x-axis indicates year, y-axis indicates BFIs, and red bars indicate 95% confidence interval.

Figure 9 shows that, despite being in the same climatic zone, different regions can exhibit differences in BFIs. For example, in the northern part of South America and the Southeast Asian region, which are both tropical karst, the BFI is significantly higher in the Southeast Asian region (81%) than in the northern part of South America (73%). There is also a significant difference in BFIs between the eastern part of the United States and the northern part of Africa, which are both arid karst climate zones. From figure 9 and figure 10 we find that BFI stability is lower and BFI values are higher in arid karst regions. The degree of variation of BFI in tropical karst regions is lower than that in arid karst regions. And subtropical and temperate karst regions have the lowest trend of base flow index change and their stability is higher.

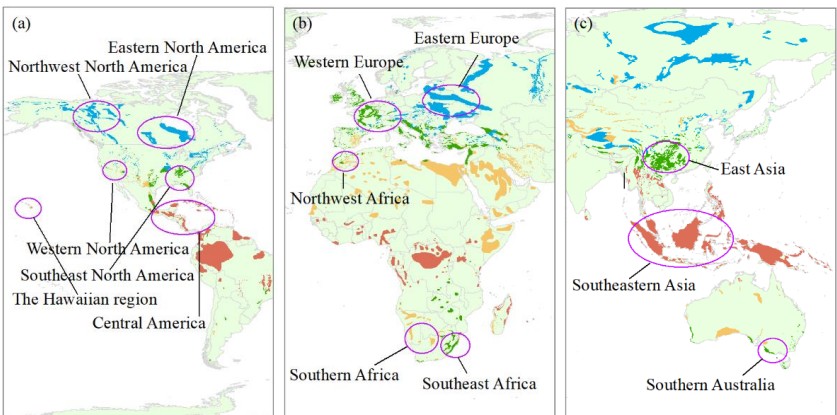

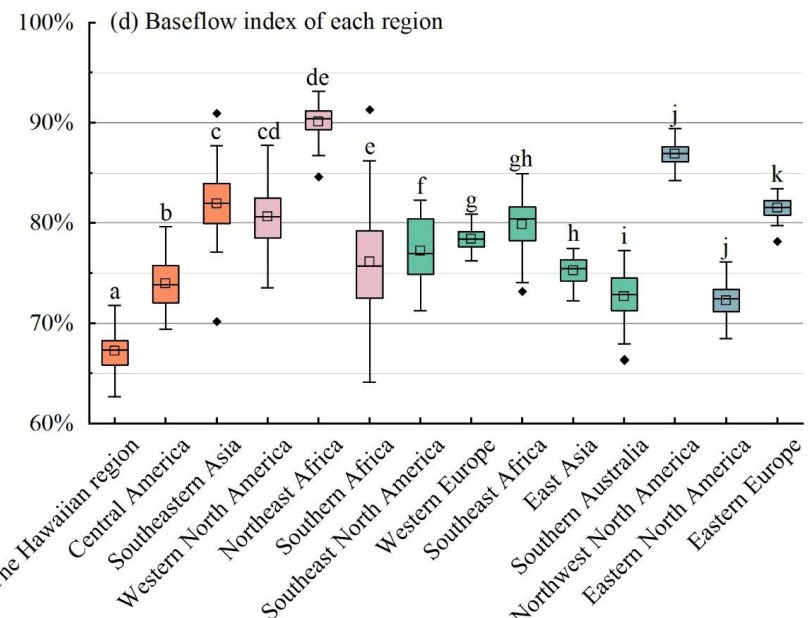


Figure 9. Distribution of BFIs in karst basins in different regions within the same
climatic zone. In figure (d), orange represents tropical karst regions, magenta represents
arid karst regions, green bars represent subtropical karst regions, and brownish-purple
represents temperate karst regions.

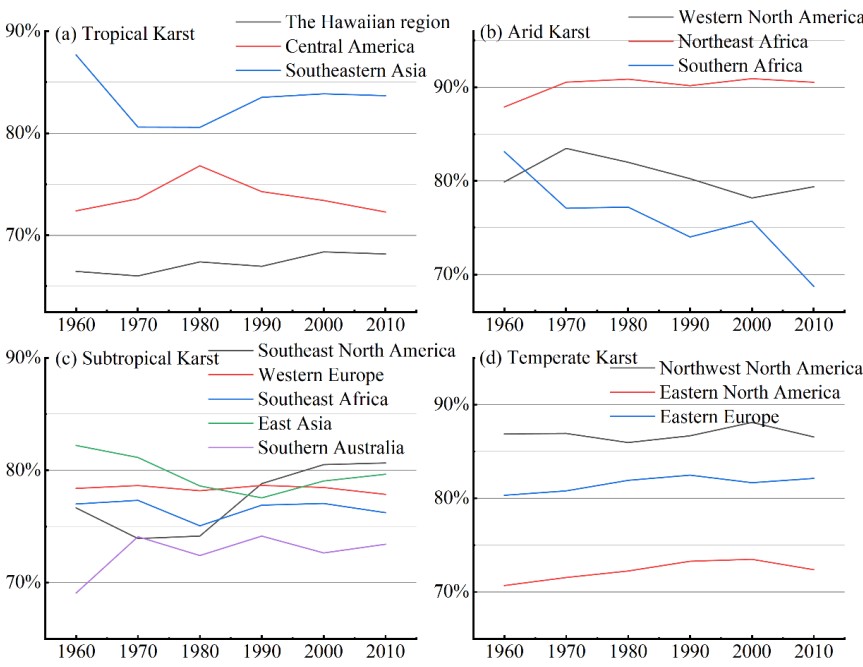




Figure 10. Characteristics of BFIs with respect to year for different regions within the
same climatic zone. Where Y-axis indicates BFI and X-axis indicates year.

## 3.4 Factors influencing baseflow indices in karst regions

Using the XGBoost model, we conducted an attribution analysis of the 12 factors
that may affect the BFI (Figure 11), and finally concluded that Gross Primary
Productivity (GPP) has the greatest influence on baseflow, reaching 15.0%. Elevation
was the next most influential factor with 12.4%, in addition to this, slope and runoff
volume (Flow) also had a large influence (>10%) on base flow index. In contrast,
climatic factors such as relative humidity (RH), Land-surface radiation (SR) and air
temperature (Temp) had a low influence on base flow index, each characterized by less
than 5%.

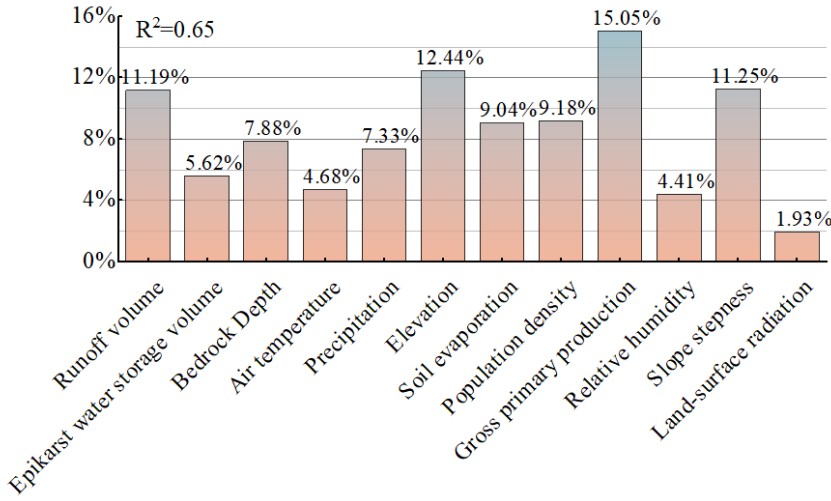

Figure 11. Influence of different factors on baseflow, where the X-axis indicates
different factors, the Y-axis indicates the contribution of the influencing factor to the
baseflow, and the number above the bar indicates the contribution of the current factor.

## 4.Discussion

## 4.1 Mechanisms of formation of baseflow characteristics in karst regions

The results of the study show that the BFI in karst regions is significantly higher
than the global average (Figure 7). We attribute this difference to the unique geological
structure and hydrological cycle characteristics of karst regions. Extensively developed
fissures, vertical seepage zones, and subsurface dissolution piping systems in karst





regions constitute complex hydrological channels, which significantly alter surface
water-groundwater exchange patterns (Ford & Williams, 2007; Li et al., 2024).
Compared with the homogeneous water storage medium dominated by fissures and
pores in non-karst areas, the network of dissolution channels in karst regions
significantly shortens the infiltration path of precipitation, and its infiltration rate can
reach several to tens of times of that in non-karst areas(Fu et al .,2016). For example,
the monitoring of karst slopes in Huanjiang, Guangxi, shows that the wet front transport
rate is as high as 1373 mm/h, compared with 17-610 mm/h in non-karst regions, which
indicates that the rate of water infiltration in karst regions is much higher than that in
non-karst regions (Medici et al., 2019; Zhang et al., 2024). This part of precipitation
recharge into the subsurface, under the action of gravity and pressure, squeezes the 'old
water' out of the underground aquifer, which indirectly enhances the baseflow ratio
(Reimann et al., 2011; Bailly-Comte et al., 2010; Evans. 1983; Ronayne. 2013). Studies
have shown that this mechanism can result in significantly higher baseflow
contributions in karst regions, even above 80% in specific environments (Zhang et al.,
2022), whereas only less than 50% of precipitation can be converted to baseflow in non-
karst regions due to the blocking effect of loose sedimentary layers (Cusano et al., 2024).

414         Significant differences in surface cover conditions further reinforce baseflow

differences. In some karst areas, bedrock is exposed to more than 60%, and thin layers
of residual soil (<30 cm) cover only 20% of the surface, a geologic feature that results
in reduced surface interception and elevated subsurface recharge (Anker et al., 2023;Li
et al .,2024 ;Wang et al .,2024). The karst fissure system is directly exposed to the
atmospheric interface, avoiding water loss through evaporation from the soil layer, and
the lack of continuous surface cover allows for direct infiltration of large amounts of
precipitation (Yang et al., 2025; Li et al., 2023). On the contrary, in non-karst areas, the
soil-vegetation system formed by thicker weathered crust constitutes a natural
evapotranspiration interface, and the average annual evapotranspiration can reach 40%
of the precipitation, and surface runoff accounts for 30% of the precipitation, which
significantly weakened the intensity of groundwater recharge (Jiang et al., 2020; Wang
et al., 2020; Wetzel et al.,1996). This double hydrological barrier effect ultimately leads
to systematic differences between BFIs in karst regions and non-karst regions.
## 4.2 Reasons for differences in baseflow in karst regions in different
climatic zones





The results show that BFIs in karst regions in different climatic zones exhibit significant differences (Figure 9 and Figure 10). The underlying driving force lies in the heterogeneity of the geologic structure and its coupling effect with long-term climatic erosion(Liu et al .,2023). Among them, the control of the spatial structure of the water storage medium by the geologic context is the decisive factor for the differences in BFIs(Luo et al ., 2023). For example, in Southeast Asian karst regions (e.g., Halong Bay, Vietnam), the development of high-purity, thick-bedded limestone, and the formation of a pipeline network with vertical dominance under the background of tectonic uplift, the short groundwater runoff paths and efficient recharge mechanisms directly enhance the baseflow (Duringer et al., 2012). In contrast, siliceous interbedding in dolomite formations in northern South America (e.g., Caatinga, Brazil) significantly increases the resistance to dissolution and reduces the connectivity of the dissolution network, a primary geologic feature that fundamentally constrains the baseflow (Teixeira et al., 2023). The intensity of tectonic activity and the stage of geomorphic evolution further strengthen regional differences. For example, strong Cenozoic uplift in Southeast Asia formed steep young landforms that promoted vertical permeability dominance. In contrast, Paleozoic stable landmasses in northern Africa (e.g., the Saharan Atlas Mountains) are dominated by horizontal cave systems, a geologic feature that also makes the baseflow in this region significantly different from other regions (Klimchouk, 2007; Jiang et al., 2020). Surface cover characteristics are equally critical as secondary geologic elements. For example, thicker soil layers in temperate zones (e.g., Slovenia) increase surface runoff diversion through delayed infiltration, whereas large areas of exposed bedrock in equatorial zones infiltrate directly through solution gaps, creating a multiplicative effect on the BFI (Li et al., 2023).

Climate elements reshape geological structures over large time scales through geological erosion processes, thereby indirectly influencing baseflow patterns. While short-term hydrological dynamics are affected by climate parameters such as precipitation intensity and seasonal distribution (Mo et al., 2021 ; Cheng et al .,2023), the profound control of climate on the baseflow index is evident in its long-term modification of karst systems. For example, the strong coupling of heavy precipitation and high temperatures in equatorial regions significantly accelerates the dissolution of carbonate rocks, forming a dense network of highly permeable dissolution fissures. Conversely, the persistent moisture associated with temperate maritime climates enhances the dissolution of carbonate rocks (with an average annual dissolution rate





approximately 40% higher than that of non-karst areas at the same latitude), leading to
the formation of cave clusters characterized by labyrinthine structures and interwoven
underground river systems. This climate-driven differentiation in dissolution alters the
capacity of groundwater storage spaces, ultimately reflecting in the characteristic values
of regional baseflow indices (Ford and Williams, 2007; Goldscheider, 2015; Tapiador
et al., 2012).

## 4.3 Reasons for changes in baseflow indices over time

The results of the experiment revealed an increasing trend in the BFI in the karst
region (Figure 8). Although the degree of increase is low (about 1.5% from 1960 to
2015), we still feel that this degree of increase in BFI is a cause for concern given that
the average BFI in the karst region is already at a high level. The reason for the
increasing trend in BFI in the karst region is presumably caused by the large loss of
groundwater. Extensive monitoring has shown that groundwater levels globally show a
rapid declining trend, and this systematic attenuation has triggered multiple crises such
as basin hydrological process anomalies and regional climate feedback imbalances
(Jasechko et al., 2024, de Graaf et al. 2019; Liu et al.,2015). It is the rapid decline of
the water table that leads to a constant unsaturated state of groundwater storage.
Therefore, when recharged by precipitation, large amounts of precipitation
preferentially fill the void in the water table, making the generation of surface runoff
require longer recharge cycles.
In addition to this, the geological and hydrological characteristics of the karst
region further amplify this effect of reduced surface runoff and increased Baseflow
(Zhu et al.,2025). On the one hand, there is the rapid water-conducting effect of the
karst fissure network, where the extensive development of dissolution pipes and
fissures in the karst bedrock accelerates vertical infiltration of precipitation into deep
groundwater, leading to difficulties in retaining soil moisture and a significant increase
in the runoff generation threshold (Hartmann et al., 2014). On the other hand, there is
the dissipative effect of the surface-subsurface dichotomy, where the thickness of the
unsaturated zone of the karst aquifer increases in the context of persistent groundwater
overdraft (D'Ettorre et al., 2024), further weakening the immediate contribution of
precipitation events to runoff.

## 4.4 Applicability and limitations of this study

With regard to data sources, the original data sources are diverse and complex.



Although we have made a lot of efforts to eliminate a large number of original
documents with distorted data and to screen out some unreasonable data in the
calculation, it is always difficult to fully balance the deficiencies in the original data.
On the boundaries of applicability of the method itself. For example, the
parameterization framework of digital filtering methods (e.g., Eckhardt and Chapman
algorithms) based on the assumption of linear recession is at variance with the nonlinear
characteristics of karst hydrological processes. Together with the rapid recession
processes dominated by karst pipe flow (rates up to 2-3 times that of porous media
basins) leads to a general underestimation of the recession coefficient (Kang et al 2022;
Rattayová & Hlavčová., 2023), which leads to differences in baseflow separation
between methods with different principles. For example, the Chapman and CM
methods used in this study, from the results of the two methods, the separation of
baseflow is significantly lower than the other methods, which is due to the lower degree
of response of the Chapman and CM methods to precipitation recharge, which is also
reflected in Helfer's study (Helfer et al., 2024). In addition, empirical parameters such
as maximum baseflow (BFI_max) are mostly derived from rate-determined results for
temperate homogeneous aquifers, and their physical mechanisms have not been fully
adapted for applicability in karst regions (Zhou et al., 2017).
Despite the above limitations, this study ensures the spatial representativeness and
methodological reliability of the study conclusions by integrating a global-scale multi-
source dataset of karst region (covering more than 85% of the typical karst
geomorphological units) and adopting standardized validation indexes (KGE, NSE).
The results show that the karst baseflow separation results can effectively characterize
the regional hydrological features and provide data support for water resource
management and eco-hydrological model construction in karst region. Future research
can integrate geophysical exploration and isotope tracer technology to develop a
dynamic parameterization scheme adapted to non-homogeneous media.
5.Conclusion
This study systematically analyzes the spatial distribution characteristics and trends
of BFIs in global karst regions. The results show that the BFI (78%) in karst regions is
generally significantly higher than the global BFI. This phenomenon confirms the
differential regulation of the runoff partitioning mechanism by the unique surface-
groundwater dichotomy in karst regions. Meanwhile, the study systematically evaluates
the applicable boundary of the hydrographic method in karst region and proves the



applicability of the hydrographic method in karst region. It is noteworthy that the BFI
in the karst region shows a phased upward trend against the background of the general
decay of global groundwater reserves. This may be related to the buffering effect of
karst aquifers on extreme climatic events and human activity-induced changes in the
subsurface bedding. In future research, we can integrate high-precision geological
tectonic data and multi-source remote sensing information to construct a coupled
climate-hydrology-geology model to quantitatively analyze the response characteristics
of hydrological fluxes of karst systems under the background of climate change, and
further improve the spatial and temporal resolution dimensions of karst water cycle
theory.



## Declaration of the Competing Interest

The authors declare that they have no known competing financial interests o r personal relationships that could have appeared to influence the work reported in this paper

## Acknowledgments

This study was supported by the National Natural Science Foundation of China (42461004, U1812401, U1612441); Science and Technology Plan Project of Guizhou Province (Qiankehejichu-ZK[2025] Zhongdian 045; Qiankehejichu-ZK[2025] Mianshang 268)

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

04817-z.

**Author contributions**
Ze Yuan: Conceptualization, Methodology, Writing – original draft.
Qiuwen Zhou: Formal analysis, Investigation, Writing – review & editing.
Yuan Li: Data curation, Visualization, Software.
Yuluan Zhao: Validation, Resources, Project administration.
Shengtian Yang: Supervision, Funding acquisition, Writing – review & editing.

**Acknowledgments**
The authors thank all those who provided valuable assistance during the experimental
process.