# Peer review of "Baseflow in karst regions is significantly higher than the"

_EGUsphere, 2025_

## Community Comment (CC2)

Thank you very much for your recognition of our article and your constructive comments and suggestions. We have carefully studied and fully understood all your suggestions, and will revise and improve the article in accordance with these suggestions to enhance its quality. The specific revision plans are as follows:

**Lines 68-71. Insert new and recent literature for fissures and pipes in karstic dolostones and limestones**
Reply: These two articles conduct research on fissures and pipes in karstic dolostones and limestones, further highlighting the particularity of karst areas. This further corroborates the view that the unique hydrological structure of karst areas makes the base flow characteristics in the region different from those in non-karst areas. Therefore, we will add citations of these two articles and their related works in the subsequent revisions.

**Lines 112-113. More detail (e.g., possible ranges and maximum value etc etc) here or in the methodology on KGE and NSE coefficients.**
Reply: For more information about the KGE and NSE coefficients, we have already provided explanations in Section 2.2.2 of the methodology. We also realize that the description of the common values and related ranges of these two coefficients lacks some details. Therefore, we will consult relevant materials to further add detailed explanations of the KGE and NSE coefficients.

**Line 114. The aim is clear, but you need to specify the 3 to 4 specific objectives of your research by using numbers (e.g., i, ii, iii).**
Reply: We will use numbers to further clarify the research objectives.

**Line 193. Here you can insert the necessary methodological details for KGE and NSE coefficients.**
Reply: Consistent with the suggestions in Lines 112-113, we will add more detailed explanations of the KGE and NSE coefficients.

**Lines 495. Four points in the discussion. Maybe, four objectives to disclose in the introduction?**
Reply: The four points in the discussion are explanations of the phenomena observed during the research process. Although we have not established a one-to-one correspondence between the citations and the discussion, generally speaking, the four points in the discussion roughly correspond to the scientific issues raised in the introduction. However, we are aware that this issue may cause confusion for readers. Therefore, we will further improve the introduction in subsequent revisions to enhance the readability and logical coherence of the article.

**Figure and tables**
Reply: Regarding the revision of figures and tables, we will make further improvements based on your suggestions combined with the actual situation to enhance their readability and standardization.

Figure 7. I would provide more detail in the introduction on the definitions of the climate. What does it mean "temperate karst". Semi-arid, humid?

Reply: The classification rules adopted here are based on the Köppen climate classification. Temperate climates are characterized by moderate conditions, without the year-round high temperatures of the tropics or the extreme cold of the polar regions, with distinct but not drastic annual temperature differences. Therefore, "temperate karst" in the figure refers to the karst areas under temperate climates. We realize that the description of this part is relatively general, which may lead readers into misunderstandings. Therefore, we will add an introduction to the Köppen climate classification in the methodology section to explain the basis for the classification used in the article.

Figure 8. Please, provide more detail on the description of the results for the (i) R², (ii) NSE and (iii) KGE values which look good. R² maybe a bit low, but this is ok.

Reply: We will add descriptions of Figure 8 in Lines 349-354, elaborating on the meanings represented by the NSE and KGE coefficients in the figure.

---

## Community Comment (CC3)

We are sincerely grateful to the reviewers for their positive assessment and the constructive suggestions provided. In accordance with their guidance, we have carefully revised the manuscript point-by-point and addressed other related issues identified during our review. The specific modifications made in response to each comment are detailed as follows.

**L27: What is the baseflow index? Please clarify!**

Reply: Let us first clarify the concept of baseflow. Baseflow refers to the sustained and stable component of total river discharge, distinct from the rapid and highly variable surface runoff driven by episodic events such as heavy rainfall or snowmelt. It is primarily sourced from slow groundwater recharge (e.g., from shallow soil moisture or deep confined aquifers). During dry seasons when precipitation recharge diminishes or ceases, baseflow becomes the critical water source that sustains river flow and supports aquatic ecosystems. The Baseflow Index (BFI) is defined as the ratio of baseflow volume to total streamflow volume over the same period. As a key metric for quantifying the extent of groundwater contribution to river discharge, the BFI often reflects the stability of the watershed's hydrological cycle.

**L36: Please provide the full name of the BFIs.**

Reply: BFI stands for "Baseflow Index." This omission occurred as we streamlined the abstract for brevity, and we acknowledge that the full term should be provided upon its first use. We will add the definition of BFI in the subsequent revision.

**L45-46: "as a slow recharge component of ..." and "as a hydrological stabiliser" wore both presented. Please rephrase this sentence.**

Reply: The sentence has been revised to: "Baseflow, as a slow recharge component from groundwater to runoff, plays a central role as a hydrological stabiliser." This new formulation accurately presents both the origin and the significance of baseflow in a logical progression, thereby addressing the lack of clarity in our initial phrasing. We have corrected this issue and will carefully scrutinize the entire manuscript to prevent similar oversights.

**L60-66: This sentence was too tedious and long. Please improve this sentence.**

Reply: We have revised this section to describe the regional differences in baseflow more concisely:" At a regional scale, BFI spatial patterns vary significantly. Studies show a higher BFI in the eastern parts of both the United States and India compared to their western regions. In China, the Yellow River Basin exhibits a pattern of high-low-high from upstream to downstream, whereas the Wei River Basin shows a gradual decline"

**L108: For the daily-scale runoff data from 1375 watersheds within the karst region, the datasets for how many years?**

Reply: The data used in this study were compiled from publicly available datasets across multiple countries, covering the period from 1880 to 2024. However, significant data gaps exist prior to 1960 and in recent years, which limits the reliability of a robust global assessment of karst baseflow characteristics. To address this, we established a screening criterion: only years with at least 500 effectively monitored basins globally were included in the analysis. The annual distribution of valid data volume is shown in the statistical subplot in the lower-right corner of Figure 1 (below). The red horizontal line in this subplot represents the threshold of 500 valid basins.

**L122: 1412 watersheds? You can directly present the 1375 stations since 37 stations were not used in present study.**

Reply: We will correct this in the manuscript to state that 1,375 data points were used. The discrepancy arose because our initial screening yielded 1,412 basins. However, during subsequent calculations, we identified that some basins suffered from severe data gaps (e.g., river flow interruption), including instances of zero recorded flow for two consecutive years. To prevent potential bias, these basins with extensive missing data were excluded, resulting in the final set of 1,375 valid data points.

186-190: Eight methods to calculate the baseflow should be described in detail. Alternatively, you can add a Table to exhibit these eight methods.

Reply: We thank the reviewer for this valuable comment. We acknowledge that the

Methodology section lacks sufficient explanation of the background and principles of each baseflow separation method used. In response, we have consolidated all the methods employed in this study into the table below, which summarizes both the background and fundamental principles of each method. To improve the manuscript's completeness, we will either incorporate detailed explanations of each method into the main text or include this table as supplementary material.

| Graphical Methods |                                                                         |  |
|-------------------|-------------------------------------------------------------------------|--|
| Name              | Description                                                             |  |
| (Abbreviation)    |                                                                         |  |
| Fixed Interval    | Proposed by Pettyjohn & Henning in 1979 and first introduced as a       |  |
| Method (FIM)      | core method within the HYSEP program. Its principle involves            |  |
|                   | segmenting the hydrograph into consecutive fixed-time intervals and     |  |
|                   | taking the minimum flow within each interval as the baseflow.           |  |
| Local Minimum     | Integrated into the HYSEP program by Sloto & Crouse in 1996 as a        |  |
| Method (LMM)      | standard graphical separation technique. The method works by            |  |
|                   | identifying local minimum points in the flow time series to demarcate   |  |
|                   | the separation between baseflow and surface runoff.                     |  |
| Sliding Window    | Also proposed by Sloto & Crouse within the HYSEP program, this          |  |
| Method (SW)       | method improves upon the rigidity of the Fixed Interval Method. Its     |  |
|                   | principle is to traverse the hydrograph using a sliding window of fixed |  |
|                   | width, dynamically calculating the minimum flow within each             |  |
|                   | window and assigning it as the baseflow value at the window's center    |  |
|                   | point.                                                                  |  |
| UK Institute of   | Originally developed by the UK Institute of Hydrology in 1980 and       |  |
| Hydrology (UKIH)  | later refined by Wels et al., who also developed its computational      |  |
|                   | program. It is a baseflow separation method that incorporates           |  |
|                   | precipitation thresholds and flow response, dynamically adjusting the   |  |
|                   | baseflow threshold to identify the separation point between rainfall    |  |
|                   | events and baseflow.                                                    |  |

| Digital Filtering Methods |                                                                       |  |
|---------------------------|-----------------------------------------------------------------------|--|
| Name                      | Description                                                           |  |
| (Abbreviation)            |                                                                       |  |
| Boughton Method           | Proposed by Boughton in 1993 as a representative single-parameter     |  |
| (Boughton)                | recursive filtering method. It calculates the baseflow at the current |  |
|                           | timestep based on the baseflow value from the previous timestep       |  |
|                           | and the total flow value at the current timestep.                     |  |
| Chapman-Maxwell           | An enhancement of the Chapman filter by Chapman & Maxwell in          |  |
| Filter Method (CM)        | 1996, which improves accuracy by dynamically adjusting the            |  |

|                      | <del>,</del>                                                          |
|----------------------|-----------------------------------------------------------------------|
|                      | recession constant. It computes baseflow by treating it as a          |
|                      | weighted average of the concurrent total streamflow and the           |
|                      | baseflow from the preceding timestep.                                 |
| Chapman Filter       | Proposed by Chapman in 1991 to address the issue of                   |
| Method (Chapman)     | unrealistically constant baseflow at the end of recession periods     |
|                      | found in the Lyne-Hollick method. Its principle involves              |
|                      | calculating baseflow as a weighted average of the current total       |
|                      | streamflow and the previous timestep's baseflow.                      |
| Exponential          | Introduced to hydrology by Vogel & Kroll in 1992 for Baseflow         |
| Weighted Moving      | Index (BFI) calculation. The method estimates baseflow by             |
| Average (EWMA)       | applying exponential weighting to smooth the streamflow time          |
|                      | series data.                                                          |
| Eckhardt Filter      | Proposed by Eckhardt in 2005, this is a two-parameter recursive       |
| Method (Eckhardt)    | filtering method. It estimates baseflow by evaluating the maximum     |
|                      | values of the recession constant and the maximum baseflow index.      |
| Furey Digital Filter | Proposed by Furey in 2001, based on a physical-statistical model of   |
| Method (Furey)       | hillslope hydrological processes. Its principle involves estimating   |
|                      | baseflow by considering the recession constant and a calibrated       |
|                      | parameter.                                                            |
| Lyne-Hollick Digital | First introduced by Lyne & Hollick in 1979, it is one of the earliest |
| Filter Method (LH)   | recursive digital filter methods. The principle involves a two-pass   |
|                      | filtering process to estimate baseflow.                               |
| Willems Digital      | Proposed by Willems in 2009, based on a linear reservoir model        |
| Filter Method        | and least squares optimization. It estimates baseflow by calculating  |
| (Willems)            | it as a weighted average of the baseflow from the previous timestep   |
|                      | and the total flow at the current timestep.                           |
|                      |                                                                       |

**2: The colors for these four karst regions were to similar. Please improve the color.**

Reply: We have adjusted the colors for the different categories to improve their distinguishability in the figure, as shown below.

**3-4: Significant difference test should be added.**

Reply: We have supplemented the significance tests for Figures 3 and 4. Using the Kruskal-Wallis test, we confirmed the statistical significance of the differences in both the KGE and NSE coefficients among the 12 baseflow separation methods. Accordingly, we will enhance the main text by adding a discussion on the performance differences between different types of methods, along with further interpretation of the effectiveness of each separation method. In the figure below, the letters denote the results of multiple comparisons based on the Mann-Whitney U test, while the horizontal lines at the bottom of the figure indicate significant differences between the graphical methods and digital filtering methods. Methods sharing the same letter show no significant difference at the p

Figure 3. Comparison of KGE coefficients (a) and NSE coefficients (b) for the 12 baseflow separation methods. The X-axis represents each separation method, and the Y-axis indicates the value of the coefficients. Green color in the plot denotes the graphical method, while orange denotes the digital filtering method. The letters above the boxes indicate significant differences among the different baseflow separation methods, while the horizontal lines in the lower part of the figure represent significant differences between the graphical method and the digital filtering method. The black line inside the boxplot denotes the mean value, with upper and lower limits set at 1.5 times the interquartile range (IQR). Values exceeding this range are considered outliers and are marked as dots at the top and bottom of the boxplot.

Figure 4. Comparison of KGE coefficients (left) and NSE coefficients (right) for karst regions in different climatic zones (as labeled in the bottom-right corner of each subplot). The X-axis represents each separation method, and the Y-axis indicates the coefficient values. The letters above the boxes indicate significant differences among the baseflow separation methods, while the horizontal lines in the lower part of the figure denote significant differences between the graphical and digital filtering method groups. Green color in the plot denotes the

graphical method, and orange represents the digital filtering method. The black line inside each boxplot indicates the mean value, with the upper and lower limits set at 1.5 times the interquartile range (IQR). Data points beyond this range are considered outliers and are marked as dots at the top and bottom of the boxplot.

**8: Please provide the P value.**

Reply: We performed the Mann-Kendall test on the data in Figure 8 using the pymannkendall library. The results reveal a statistically significant increasing trend in the baseflow characteristics, with a p-value of 0.00002 (

---

## Community Comment (CC5)

Thank you for your recognition of our manuscript and your valuable comments. Below are the detailed responses to each comment.

**1. The baseflow separation methods seem to generate distinct BFIs and event contrasting trends in certain time periods. How do you tackle the different separation results?**

Reply: Regarding the discrepancies between baseflow results derived from different separation methods, our core criterion for evaluation is to compare the Nash-Sutcliffe Efficiency (NSE) and Kling-Gupta Efficiency (KGE) coefficients between each method's output and the source data. Notably, the NSE and KGE coefficients we calculated were specific to runoff periods excluding those with precipitation recharge. During these periods, river discharge is solely replenished by baseflow—thus, there should be a clear consistency between total river discharge and baseflow volume.

However, after precipitation recharge ceases, some surface runoff may still be en route to the river. This means river discharge during this phase is sourced not only from baseflow but also from residual surface runoff. To mitigate this issue, we specifically selected small watersheds with an area of less than 2,500 square kilometers. For such small watersheds, the travel time of surface runoff to the river is relatively short, which further minimizes the impact of surface runoff on the accuracy of baseflow separation results.

Thus, we believe using these two coefficients is a viable approach for assessing the reliability of baseflow separation methods. The specific assessment methods and the process for excluding precipitation recharge periods can be found in the Methods section of this manuscript.

**2. In Figure 7, the authors compared BFI in karst regions and the global average. However, the global average is impacted by the coverage ratio of karst regions. Consider replace the global mean with non-karst area mean.**

Reply: Your comment is critical, and we will revise Figure 7 in the revised version of the manuscript to better achieve the effect of comparison with global-scale data. Specifically, we chose the mean baseflow curve of global non-karst regions as the comparison dataset for the karst baseflow curve in Figure 7. This choice is based on the following consideration: the number of catchment datasets for global non-karst regions exceeds 6,000, while the dataset used in this study to calculate karst baseflow values comprises 1,375 catchments. Due to this difference in dataset size, the difference between the global average baseflow curve and the global non-karst average baseflow curve is minimal (less than 1%). We therefore opted to use the global non-karst baseflow curve as the comparison dataset.

**3. Figure 11 shows the factor impact on BFI, but this does not reveal if the impact is positive or negative. Consider including analysis such as SHAP.**

Reply: Your consideration is valid. Initially, we only focused on the magnitude of the impact of different factors on the Base Flow Index (BFI), without fully incorporating their impact characteristics. In the revised version of the study, we will incorporate SHAP values (SHapley Additive exPlanations) to refine and supplement the analysis of how different factors influence the BFI, including their specific impact characteristics.

---

## Author Comment (AC1)

We are sincerely grateful to the reviewers for their positive assessment and the constructive suggestions provided. In accordance with their guidance, we have carefully revised the manuscript point-by-point and addressed other related issues identified during our review. The specific modifications made in response to each comment are detailed as follows.

**L27: What is the baseflow index? Please clarify!**

Reply: Let us first clarify the concept of baseflow. Baseflow refers to the sustained and stable component of total river discharge, distinct from the rapid and highly variable surface runoff driven by episodic events such as heavy rainfall or snowmelt. It is primarily sourced from slow groundwater recharge (e.g., from shallow soil moisture or deep confined aquifers). During dry seasons when precipitation recharge diminishes or ceases, baseflow becomes the critical water source that sustains river flow and supports aquatic ecosystems. The Baseflow Index (BFI) is defined as the ratio of baseflow volume to total streamflow

The Baseflow Index (BFI) is defined as the ratio of baseflow volume to total streamflow volume over the same period. As a key metric for quantifying the extent of groundwater contribution to river discharge, the BFI often reflects the stability of the watershed's hydrological cycle.

We recognize that the explanation of baseflow was not sufficiently elaborated, which has caused confusion among readers and reviewers. Therefore, in the subsequent revision, we will enhance the description of baseflow in both the abstract and introduction to more clearly articulate the research focus of the paper.

**L36: Please provide the full name of the BFIs.**

Reply: BFI stands for "Baseflow Index." This omission occurred as we streamlined the abstract for brevity, and we acknowledge that the full term should be provided upon its first use. We will add the definition of BFI in the subsequent revision.

**L45-46: "as a slow recharge component of ..." and "as a hydrological stabiliser" wore both presented. Please rephrase this sentence.**

Reply: The sentence has been revised to: "Baseflow, as a slow recharge component from groundwater to runoff, plays a central role as a hydrological stabiliser." This new formulation accurately presents both the origin and the significance of baseflow in a logical progression, thereby addressing the lack of clarity in our initial phrasing. We have corrected this issue and will carefully scrutinize the entire manuscript to prevent similar oversights.

**L60-66: This sentence was too tedious and long. Please improve this sentence.**

Reply: We have revised this section to describe the regional differences in baseflow more concisely:" At a regional scale, BFI spatial patterns vary significantly. Studies show a higher

BFI in the eastern parts of both the United States and India compared to their western regions. In China, the Yellow River Basin exhibits a pattern of high-low-high from upstream to downstream, whereas the Wei River Basin shows a gradual decline"

**L108: For the daily-scale runoff data from 1375 watersheds within the karst region, the datasets for how many years?**

Reply: The data used in this study were compiled from publicly available datasets across multiple countries, covering the period from 1880 to 2024. However, significant data gaps exist prior to 1960 and in recent years, which limits the reliability of a robust global assessment of karst baseflow characteristics. To address this, we established a screening criterion: only years with at least 500 effectively monitored basins globally were included in the analysis. The annual distribution of valid data volume is shown in the statistical subplot in the lower-right corner of Figure 1 (below). The red horizontal line in this subplot represents the threshold of 500 valid basins.

**L122: 1412 watersheds? You can directly present the 1375 stations since 37 stations were not used in present study.**

Reply: We will correct this in the manuscript to state that 1,375 hydrological stations were used. The discrepancy arose because our initial screening yielded 1,412 basins. However, during subsequent calculations, we identified that some basins suffered from severe data gaps (e.g., river flow interruption), including instances of zero recorded flow for two consecutive years. To prevent potential bias, these basins with extensive missing data were excluded, resulting in the final set of 1,375 valid data points.

186-190: Eight methods to calculate the baseflow should be described in detail. Alternatively, you can add a Table to exhibit these eight methods.

Reply: We thank the reviewer for this valuable comment. We acknowledge that the Methodology section lacks sufficient explanation of the background and principles of each baseflow separation method used. In response, we have consolidated all the methods employed in this study into the table below, which summarizes both the background and fundamental principles of each method. To improve the manuscript's completeness, we will either incorporate detailed explanations of each method into the main text or include this table as supplementary material.

| Graphical Methods |                                                                         |  |
|-------------------|-------------------------------------------------------------------------|--|
| Name              | Description                                                             |  |
| (Abbreviation)    |                                                                         |  |
| Fixed Interval    | Proposed by Pettyjohn & Henning in 1979 and first introduced as a       |  |
| Method (FIM)      | core method within the HYSEP program. Its principle involves            |  |
|                   | segmenting the hydrograph into consecutive fixed-time intervals and     |  |
|                   | taking the minimum flow within each interval as the baseflow.           |  |
| Local Minimum     | Integrated into the HYSEP program by Sloto & Crouse in 1996 as a        |  |
| Method (LMM)      | standard graphical separation technique. The method works by            |  |
|                   | identifying local minimum points in the flow time series to demarcate   |  |
|                   | the separation between baseflow and surface runoff.                     |  |
| Sliding Window    | Also proposed by Sloto & Crouse within the HYSEP program, this          |  |
| Method (SW)       | method improves upon the rigidity of the Fixed Interval Method. Its     |  |
|                   | principle is to traverse the hydrograph using a sliding window of fixed |  |
|                   | width, dynamically calculating the minimum flow within each             |  |
|                   | window and assigning it as the baseflow value at the window's center    |  |
|                   | point.                                                                  |  |
| UK Institute of   | Originally developed by the UK Institute of Hydrology in 1980 and       |  |
| Hydrology (UKIH)  | later refined by Wels et al., who also developed its computational      |  |
|                   | program. It is a baseflow separation method that incorporates           |  |
|                   | precipitation thresholds and flow response, dynamically adjusting the   |  |
|                   | baseflow threshold to identify the separation point between rainfall    |  |
|                   | events and baseflow.                                                    |  |

|                 | Digital Filtering Methods                                             |
|-----------------|-----------------------------------------------------------------------|
| Name            | Description                                                           |
| (Abbreviation)  |                                                                       |
| Boughton Method | Proposed by Boughton in 1993 as a representative single-parameter     |
| (Boughton)      | recursive filtering method. It calculates the baseflow at the current |

| timestep based on the baseflow value from the previous timestep and the total flow value at the current timestep.  Chapman-Maxwell Filter Method (CM)  An enhancement of the Chapman filter by Chapman & Maxwell 1996, which improves accuracy by dynamically adjusting the recession constant. It computes baseflow by treating it as a weighted average of the concurrent total streamflow and the baseflow from the preceding timestep.  Chapman Filter  Proposed by Chapman in 1991 to address the issue of |    |
|-----------------------------------------------------------------------------------------------------------------------------------------------------------------------------------------------------------------------------------------------------------------------------------------------------------------------------------------------------------------------------------------------------------------------------------------------------------------------------------------------------------------|----|
| Chapman-Maxwell Filter Method (CM)  An enhancement of the Chapman filter by Chapman & Maxwell 1996, which improves accuracy by dynamically adjusting the recession constant. It computes baseflow by treating it as a weighted average of the concurrent total streamflow and the baseflow from the preceding timestep.                                                                                                                                                                                         |    |
| Filter Method (CM)  1996, which improves accuracy by dynamically adjusting the recession constant. It computes baseflow by treating it as a weighted average of the concurrent total streamflow and the baseflow from the preceding timestep.                                                                                                                                                                                                                                                                   |    |
| recession constant. It computes baseflow by treating it as a weighted average of the concurrent total streamflow and the baseflow from the preceding timestep.                                                                                                                                                                                                                                                                                                                                                  | in |
| weighted average of the concurrent total streamflow and the baseflow from the preceding timestep.                                                                                                                                                                                                                                                                                                                                                                                                               |    |
| baseflow from the preceding timestep.                                                                                                                                                                                                                                                                                                                                                                                                                                                                           |    |
|                                                                                                                                                                                                                                                                                                                                                                                                                                                                                                                 |    |
| Chapman Filter Proposed by Chapman in 1991 to address the issue of                                                                                                                                                                                                                                                                                                                                                                                                                                              |    |
| Chapman in 1991 to address the issue of                                                                                                                                                                                                                                                                                                                                                                                                                                                                         |    |
| Method (Chapman) unrealistically constant baseflow at the end of recession periods                                                                                                                                                                                                                                                                                                                                                                                                                              |    |
| found in the Lyne-Hollick method. Its principle involves                                                                                                                                                                                                                                                                                                                                                                                                                                                        |    |
| calculating baseflow as a weighted average of the current total                                                                                                                                                                                                                                                                                                                                                                                                                                                 |    |
| streamflow and the previous timestep's baseflow.                                                                                                                                                                                                                                                                                                                                                                                                                                                                |    |
| Exponential Introduced to hydrology by Vogel & Kroll in 1992 for Baseflow                                                                                                                                                                                                                                                                                                                                                                                                                                       |    |
| Weighted Moving Index (BFI) calculation. The method estimates baseflow by                                                                                                                                                                                                                                                                                                                                                                                                                                       |    |
| Average (EWMA) applying exponential weighting to smooth the streamflow time                                                                                                                                                                                                                                                                                                                                                                                                                                     |    |
| series data.                                                                                                                                                                                                                                                                                                                                                                                                                                                                                                    |    |
| Eckhardt Filter Proposed by Eckhardt in 2005, this is a two-parameter recursive                                                                                                                                                                                                                                                                                                                                                                                                                                 |    |
| Method (Eckhardt)   filtering method. It estimates baseflow by evaluating the maximu                                                                                                                                                                                                                                                                                                                                                                                                                            | m  |
| values of the recession constant and the maximum baseflow inde                                                                                                                                                                                                                                                                                                                                                                                                                                                  | x. |
| Furey Digital Filter   Proposed by Furey in 2001, based on a physical-statistical model                                                                                                                                                                                                                                                                                                                                                                                                                         | of |
| Method (Furey) hillslope hydrological processes. Its principle involves estimating                                                                                                                                                                                                                                                                                                                                                                                                                              | ,  |
| baseflow by considering the recession constant and a calibrated                                                                                                                                                                                                                                                                                                                                                                                                                                                 |    |
| parameter.                                                                                                                                                                                                                                                                                                                                                                                                                                                                                                      |    |
| Lyne-Hollick Digital   First introduced by Lyne & Hollick in 1979, it is one of the earlie                                                                                                                                                                                                                                                                                                                                                                                                                      | st |
| Filter Method (LH)   recursive digital filter methods. The principle involves a two-pass                                                                                                                                                                                                                                                                                                                                                                                                                        | S  |
| filtering process to estimate baseflow.                                                                                                                                                                                                                                                                                                                                                                                                                                                                         |    |
| Willems Digital Proposed by Willems in 2009, based on a linear reservoir model                                                                                                                                                                                                                                                                                                                                                                                                                                  |    |
| Filter Method and least squares optimization. It estimates baseflow by calculating                                                                                                                                                                                                                                                                                                                                                                                                                              | ng |
| (Willems) it as a weighted average of the baseflow from the previous timest                                                                                                                                                                                                                                                                                                                                                                                                                                     | ep |
| and the total flow at the current timestep.                                                                                                                                                                                                                                                                                                                                                                                                                                                                     |    |

**2: The colors for these four karst regions were to similar. Please improve the color.**

Reply: We have adjusted the colors for the different categories to improve their distinguishability in the figure, as shown below.

**3-4: Significant difference test should be added.**

Reply: We have supplemented the significance tests for Figures 3 and 4. Using the Kruskal-Wallis test, we confirmed the statistical significance of the differences in both the KGE and NSE coefficients among the 12 baseflow separation methods. Accordingly, we will enhance the main text by adding a discussion on the performance differences between different types of methods, along with further interpretation of the effectiveness of each separation method. In the figure below, the letters denote the results of multiple comparisons based on the Mann-Whitney U test, while the horizontal lines at the bottom of the figure indicate significant differences between the graphical methods and digital filtering methods. Methods sharing the same letter show no significant difference at the p

Figure 3. Comparison of KGE coefficients (a) and NSE coefficients (b) for the 12 baseflow separation methods. The X-axis represents each separation method, and the Y-axis indicates the value of the coefficients. Green color in the plot denotes the graphical method, while orange denotes the digital filtering method. The letters above the boxes indicate significant differences among the different baseflow separation methods, while the horizontal lines in the lower part of the figure represent significant differences between the graphical method and the digital filtering method. The black line inside the boxplot denotes the mean value, with upper and lower limits set at 1.5 times the interquartile range (IQR). Values exceeding this range are considered outliers and are marked as dots at the top and bottom of the boxplot.

Figure 4. Comparison of KGE coefficients (left) and NSE coefficients (right) for karst regions in different climatic zones (as labeled in the bottom-right corner of each subplot). The X-axis represents each separation method, and the Y-axis indicates the coefficient values. The letters above the boxes indicate significant differences among the baseflow separation methods, while the horizontal lines in the lower part of the figure denote significant differences between the graphical and digital filtering method groups. Green color in the plot denotes the

graphical method, and orange represents the digital filtering method. The black line inside each boxplot indicates the mean value, with the upper and lower limits set at 1.5 times the interquartile range (IQR). Data points beyond this range are considered outliers and are marked as dots at the top and bottom of the boxplot.

**8: Please provide the P value.**

Reply: We performed the Mann-Kendall test on the data in Figure 8 using the pymannkendall library. The results reveal a statistically significant increasing trend in the baseflow characteristics, with a p-value of 0.00002 (